# Application Prospects of MXenes Materials Modifications for Sensors

**DOI:** 10.3390/mi14020247

**Published:** 2023-01-18

**Authors:** Vy Anh Tran, Nguyen Tien Tran, Van Dat Doan, Thanh-Quang Nguyen, Hai Ha Pham Thi, Giang N. L. Vo

**Affiliations:** 1Institute of Applied Technology and Sustainable Development, Nguyen Tat Thanh University, Ho Chi Minh City 700000, Vietnam; 2Faculty of Environmental and Food Engineering, Nguyen Tat Thanh University, Ho Chi Minh City 700000, Vietnam; 3Center for Advanced Chemistry, Institute of Research and Development, Duy Tan University, 03 Quang Trung, Da Nang 550000, Vietnam; 4Faculty of Natural Sciences, Duy Tan University, 03 Quang Trung, Da Nang 550000, Vietnam; 5The Faculty of Chemical Engineering, Industrial University of Ho Chi Minh City, Ho Chi Minh City 700000, Vietnam; 6Department of External Relations and Project Development, Institute of Applied Science and Technology (IAST), Van Lang University, Ho Chi Minh City 700000, Vietnam; 7NTT Hi-Tech Institute, Nguyen Tat Thanh University, 298-300A Nguyen Tat Thanh Street, Ward 13, District 4, Ho Chi Minh City 700000, Vietnam; 8Faculty of Pharmacy, University of Medicine and Pharmacy at Ho Chi Minh City, Ho Chi Minh City 700000, Vietnam

**Keywords:** MXenes nanomaterial, biosensor, electrochemistry sensor, gas sensor, optical sensor, wearable sensors

## Abstract

The first two-dimensional (2D) substance sparked a boom in research since this type of material showed potential promise for applications in field sensors. A class of 2D transition metal nitrides, carbides, and carbonitrides are referred to as MXenes. Following the 2011 synthesis of Ti_3_C_2_ from Ti_3_AlC_2_, much research has been published. Since these materials have several advantages over conventional 2D materials, they have been extensively researched, synthesized, and studied by many research organizations. To give readers a general understanding of these well-liked materials, this review examines the structures of MXenes, discusses various synthesis procedures, and analyzes physicochemistry properties, particularly optical, electronic, structural, and mechanical properties. The focus of this review is the analysis of modern advancements in the development of MXene-based sensors, including electrochemical sensors, gas sensors, biosensors, optical sensors, and wearable sensors. Finally, the opportunities and challenges for further study on the creation of MXenes-based sensors are discussed.

## 1. Introduction

Early transition metal carbonitrides and carbides in two dimensions, known as MXenes, have become a distinctive class of metallic materials with a layered structure and appealing properties, including conductivity comparable to that of metals, improved ionic conductivity, and hydrophilicity resulting from their mechanical flexibility and surfaces that are hydroxylated or oxygenated [1]. The development of single-layer, multi-layer, or nanoparticle nanosheets, which have significant surface areas and are advantageous for improving the performance of MXenes-based sensors, may be successfully regulated using tunable etching techniques [2,3].

Some classical and modern nanomaterials are applied in sensors and related fields, such as Silica NPs [4,5], Silicon NPs [6], MOF nanomaterials [7,8,9], and oxide metal NPs [10,11], although they still have certain limitations. Due to their interesting features and predictable fabrication processes, MXenes, a new two-dimensional (2D) material, were demonstrated to be promising in wearable sensory applications in 2011. MXenes are a class of 2D early transition nitride, metal carbides, and carbonitrides that are created by the group that selectively etches IIIA/IVA elements from three-dimensional (3D) MAX -phases [12]. The M_n+1_AX_n_ layers (n equals 1, 2, or 3) composition, where “M” is an abbreviation for the early transition metal of Mo, Ti, Cr, Nb, etc., and “X” is carbon/nitrogen that is associated layers of atoms in A, named from the major group element, indicate the existence of the 3D MAX phases (group IIIA or IVA). MXenes, a rising star of 2D materials, are unique in that they combine the hydrophilicity of their terminating surfaces with the metallic conductivity of transition metal carbonitrides or carbides [12,13]. Because of its unusual accordion-like structure and the functional groups that are added to the surface during their synthesis, MXenes have enticing electrical, optical, and magnetic properties that can be used for sensing, energy storage, and electromagnetic shielding [1,2,3].

Due to the rapid development of information technologies and electronics, such as wearable electronics, the internet of things, home automation, electromagnetic shielding, and intelligent industries, sensors are becoming increasingly important in our daily lives (Figure 1). There is no question that sensing materials have the greatest influence on how well they operate. With their adaptable surface characteristics, tunable bandgap, and superior mechanical strength, MXenes, a developing 2D material, are appealing in a variety of applications [13,14]. Here, we specifically concentrate on recent developments in MXene-based sensor research, discuss the advantages of MXenes and its derivatives as signal-collection materials, and attempt to clarify the design concepts and operation of the corresponding MXene-based sensors, such as electrochemical sensors, wearable sensors, gas sensors, biosensors, and optical sensors. We discuss the main issues and prospects for future MXene-based materials in sensor applications.

## 2. Synthesis and Classification of MXenes Structure

### 2.1. Synthesis of MXenes

MXenes can be made using one of two processes. The top-down approach can be used to exfoliate multiple layers of materials to create a single or few-layer MXenes sheet. The creation of MXenes from molecules/atoms is the emphasis of the second method, which is a bottom-up strategy [15,16].

#### 2.1.1. Top-Down Method

Strong covalent connections in the MAX phase between the A and MX layers are broken down by selective etching. The main technique is etching with molten salts and hydrofluoric acid (HF). The M-A strong metal connection is replaced in this process by hydroxyl, oxygen, and fluorine. Etching and exfoliation are the two primary phases in the HF process to obtain 2D MXenes. Although the direct application of HF is simple and useful, it harms the environment and human health. A fluorinated salt can be converted into in-situ HF by reacting it with a weak acid that is less harmful to MXene surfaces [16,17].

Fluorinated Lewis salts, or molten salts, are used in the molten salt method. Fluoride is not used in the synthesis, which lowers the danger involved. The synthesis of MXenes in molten salts follows a similar procedure to traditional HF techniques (Figure 2a): High-temperature molten salts of ZnCl_2_ and CuCl_2_ strip a wider variety of MAX phase materials. Zn^2+^, Cu^2+^, and Cl are consistent with acting F^-^ and H^+^ in HF in the molten salt by Lewis acids [18]. For MAX etching, electrochemical etching and minimally intense layer delamination (MILD) can also be employed to create high-quality, non-toxic MXenes [19,20].

#### 2.1.2. Bottom-Up Method

Methods for bottom-up synthesis, including plasma-enhanced pulsed laser deposition, template-based synthesis, and chemical vapor deposition, have been documented. This approach results in MXenes with a good crystalline quality, controlled structure, and size [21,22].

The stability of MXenes is a crucial characteristic that restricts its use to some extent. Its stability has been improved by researchers. Relatively mild reaction conditions are required since high HF concentrations hasten MXenes’ breakdown and alter their structure [23]. The oxidation of MXenes is reduced by organic solvents. To stop oxidation, contact with water should be limited as much as possible. MXenes oxidize more quickly in liquid than in solid environments, and photocatalysis and thermocatalysis speed up this breakdown process [24,25].

**Figure 2 micromachines-14-00247-f002:**
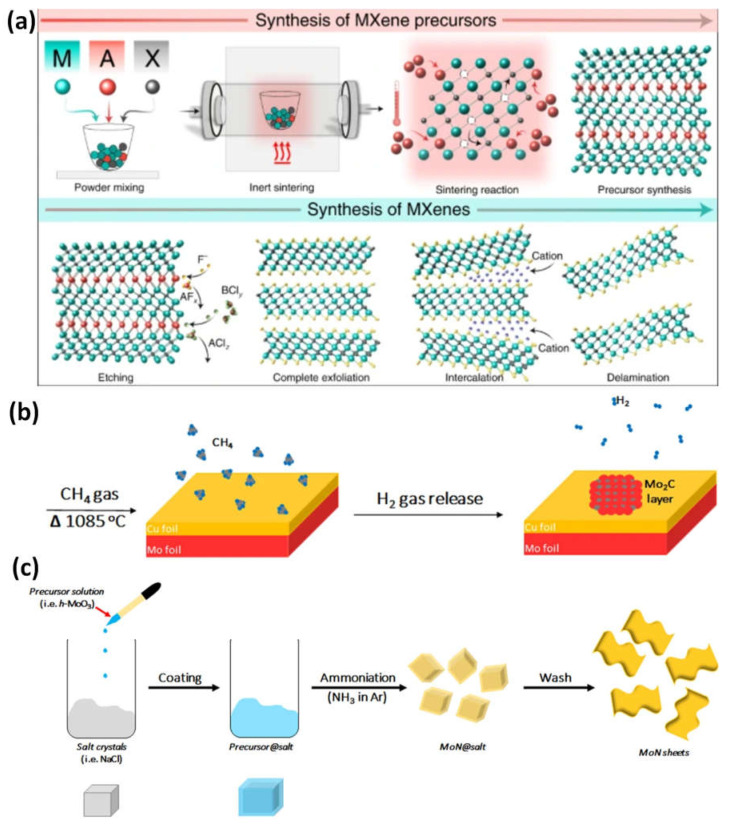
(**a**) Steps in the synthesis of MXene nanomaterials by top-down method from precursor to etching; bottom-up techniques are shown schematically in (**b**) chemical vapor deposition of Mo and C to generate a thin film of Mo_2_C in a gas chamber and (**c**) the synthesis of MoN nanosheets using a salt template. Reproduced with permission from references [18,26,27].

The initial basic elements for bottom-up synthesis are often small organic or inorganic molecules or atoms, which are then followed by crystal formation that can be organized to generate a 2D-ordered layer. The most popular technology for this strategy is chemical vapor deposition, which can create superior thin films on various substrates (Figure 2b). Utilizing methane gas as the carbon source and a Cu/Mo foil as the substrate, high-quality ultrathin Mo_2_C was created using the chemical vapor deposition method for MXene at temperatures over 1085 °C (Figure 2c). Various films with lateral sizes of approximately 50 µm were developed by optimizing the growth temperature and duration. The Mo_2_C films created were flawless and had a high degree of crystallinity, which may point to without surface functional groups. Due to well-functionalized MXenes being desired for the surface engineering of the nanosheets and the size, m, being too large for cell invasion, this is regrettably not the best solution for biomedical applications. In addition to chemical vapor deposition, the synthesis of MXene has also been investigated using the template approach and pulsed laser deposition with plasma enhancement [23].

### 2.2. Classification of MXenes Structure 

#### 2.2.1. Mono Transition Metal MXenes

As inherited from the parent MAX phases, MXenes adopt three configurations with one metal on the M site: M_4_C_3_, M_3_C_2_, and M_2_C. They are created by carefully MAX phase etching out the A element or other layered precursor, such as Mo_2_Ga_2_C, with the generic formula M_n+1_AX_n_, where X is C or N (n = 1–4), A is an element from groups 13 or 14, and M is an early transition metal. The P63/mmc layered hexagonal structure of MAX phases features virtually closed-packed M layers, and X atoms occupy octahedral positions (Figure 3a). As a result, the A element, which is metallically bound to the M element, is interleaved with M_n+1_X_n_ layers [28].

#### 2.2.2. Double Transition Metal MXenes

As opposed to their mono-transition-metal counterparts, the double transition-metal (DTM) MXenes allow TDM to possess the metal sites. Transition metals are arranged in either an in-plane or an out-of-plane ordered framework in ordered DTM MXenes. Some DTM MXenes also exist as random crystalline solids, which are specified by two transition metals that are randomly positioned across the 2D structure (Figure 3b). It is possible to tailor DTM MXenes for certain magnetic, optical, thermoelectric, electrochemical, mechanical, and catalytic properties thanks to their diverse structural makeup and variety of transition-metal pairings. This level of ability to influence their composition and structure is exceptional in real 2D materials and opens up a new route for the development of useful nanoparticles based on specific applications [29,30].

## 3. Physicochemical Properties of MXenes Materials

MXenes belong to a wide and flexible surface interplay because of the abundance of electrons linked to transition metal atoms. Raw materials, the MAX phase, etching, and delamination techniques are responsible for controlling the intriguing features of MXenes, such as their magnetic, electrical, physical, mechanical, chemical, thermal, redox, and surface capabilities. Additionally, by adjusting the composition, choosing different X and M elements, and modifying the surface terminal’s functions, the properties of MXenes can be altered for the right applications [31,32].

### 3.1. Optical Properties

Theoretically, the estimation of the imaginary part of the dielectric function tensor, which is a function of photon wavelength, can be used to examine numerous optical properties, such as absorption, reflection, and transmittance. In addition to bare structures, the optical characteristics of functionalized Ti_2_C and Ti_3_C with –O, –OH, and –F surface terminations were theoretically studied. MXenes with –F and –OH functionalization show lower in-plane absorption coefficients in a range of ultraviolet to infrared light than those without any functionalization. Ti_2_CT_2_ and Ti_3_C_2_T_2_ with –F and –OH terminals should show a white color in complement [33,34].

In an aqueous solution, MXenes exhibit a weak photoluminescence (PL) response. Making quantum dots (QDs) will go a long way toward increasing its application possibilities, particularly in the medical and optical areas. This is due to the possibility that when atomically thin QDs are created, quantum confinement and edge effects may appear. Contrary to thin and wide nanosheets, quantum dots are incredibly tiny and luminescent. By hydrothermally cutting the bulk-layered Ti_3_C_2_ MXene, MQDs can be produced with up to 10% quantum yields. It was possible to produce colloidal MQDs with various morphologies by adjusting the reaction temperature [35,36].

### 3.2. Electronic Properties

To conduct a thorough investigation of the band structures and many other electronic features, various studies have been conducted. As with MAX phases, all of the Fermi energy in bare MXenes is provided by the d orbital of the M element. With a very small band gap, the p band of the X element is situated just below the d band. Additional research revealed that strain or an outside electric field could significantly affect the band gap of MXene. Comparing MXenes to other 2D materials like graphene, their electrical properties set them apart from the others (Table 1). Recent studies have shown that the primary element influencing the electrical characteristics of MXene is the metal layers (M-layer). It is established that a fine balance of temperature and etchant activity must be kept and that a 2D carbide’s two outer transition M-layers may be changed to control the electrical transition from metallic to semiconductor-like characteristics.

Because of surface termination and varying degrees of imperfections, fabrication techniques can also affect electroconductibility. Because of the absence of organized structure by the free passage of electrons, MXenes have low electroconductivity and more flaws. Different atoms can be doped to regulate defects. Charge imbalance defects will be created through the ion’s interchange with various densities of electricity. MXenes typically have –F, –O, and –OH functional groups on their surface because fluoride makes up the majority of the etching solution. The electrical conductivity of MXenes altered in line with the surface termination [2].

### 3.3. Structural Properties

MXenes with good structural and chemical stability can serve as carriers of intrinsically active compounds and functional materials for storage, energy conversion, and sensors because of their superior conductivity and rich surface functions. Contrarily, MXenes exhibit the disadvantages of an ultrathin 2D material, such as a good tendency to restack and the absence of a constrained porous structure. Porous MXenes have undergone extensive design and modification during the past three years. To date, several suitable topologies for porous MXenes have been created utilizing synthetic processes and used in many applications, all of which have shown improved performances [44].

### 3.4. Mechanical Properties

Based on surface terminations, MXenes’ mechanical properties might vary greatly. By eliminating their charge, The Ti-C bond is weakened by the surface terminal groups. The longer Ti-C bond in Ti_2_C-Tx compared to Ti_3_C_2_-Tx could have an impact on the material’s ability to stretch. The researchers also suggested that altering the binding strength might aid in enhancing flexibility. It has been suggested that MXenes with an activity of O-terminal are the primary choice for supercapacitors, structural materials, and many applications due to their increased mechanical strength [45].

Ti_3_C_2_T_x_ has been comprehensively researched for some applications because of its mechanical capabilities, superior conductivity, and high electromagnetic shielding effect, despite the major advancements in manufacturing MXene nanostructures. In multiple studies, MXenes have shown significant mechanical ion adsorption capacities, opening the door for further inquiry into their potential use in flexible electronics and sensors. The hydrophilic properties of 2D transition metal carbonitrides and carbides, as well as their remarkable superior conductivity and surface area, led to their outstanding performance in the electromagnetic interference (EMI) shielding industry. Through the use of a sol-gel process, Ti_3_C_2_T_x_ MXene was combined with a formaldehyde and resorcinol solution to create a hybrid gel and Ti_3_C_2_T_x_ MXene/C hybrid foam (MCF). MCF/epoxy EMI-shielding composites were created using vacuum-assisted impregnation and curing techniques [46,47].

## 4. Application of MXenes Material for Sensors

### 4.1. Application of MXene in Gas Sensors

The surface electric state of MXenes changes as a result of the adsorption/desorption process, and active flaws on the surface of MXenes may be where gas absorption occurs as a result of interaction with functional surfaces [48,49]. Due to the modest intermolecular force with functional groups, electrostatic gas adsorption causes only minor resistance changes [50,51]. Gas molecules’ replacement of surface functional groups may potentially be a cause of gas absorption. This can result in carrier transfer between the adsorbent and adsorbate gas as well as major changes in the material’s resistance [2,13,52].

A simple surface-treating and exfoliation-based method was used to create Ti_3_C_2_T_x_/WSe_2_ nanohybrids, which can be used to detect various VOCs at room temperature. Ti_3_C_2_T_x_/WSe_2_ sensors can be repeatedly made via inkjet printing, which results in minimal device-to-device variance and reproducible sensing measures (Figure 4a–c). The many heterojunction interfaces created by Ti_3_C_2_T_x_/WSe_2_ nanohybrids are likely what improved the sensor performance to oxygen-containing volatile organic compounds (VOCs). The potential for using flexible sensors as useful gas-sensing IoT devices is very strong [49]. For early disease detection, VOC detection at the ppb level is essential. Existing sensor materials are unable to concurrently satisfy the requirements of strong signals and low electrical noise to obtain high sensitivity. The sensitivity of traditional semiconductor channel materials is vastly outperformed by 2D metal carbide MXenes, which have a fully functionalized surface for a strong signal and good metallic conductivity for low noise (Figure 4d). At ambient temperature, Ti_3_C_2_T_x_ MXene gas sensors demonstrated a detection limit of 50–100 ppb for VOC gases [53,54].

The very precise identification and detection of various VOCs and the concentration estimation of the target VOCs in varying backgrounds were made possible by a Ti_3_C_2_Tx-based virtual sensor array (VSA). Without requiring any temperature changes, the VSA’s reactions to the spectrum of bandwidth impedance produce a distinct fingerprint of VOCs (Figure 5a). With a detection accuracy of 93.2% for ethanol in the presence of various concentrations in water, this device exhibits highly specific identifications for various types of VOC and mixtures based on linear discrimination analysis and principal component analysis techniques [54]. Density functional theory was used to determine the NH_3_ adsorption energies on the surfaces of MXenes (M_2_C, M = Cr, and Fe), as well as their oxygen-functionalized counterparts (O-MXenes or M_2_CO_2_). The on-top sites for NH_3_ adsorption on M_2_C and M_2_CO_2_ have been initiated, according to DFT-D4 calculations. The E_a_ is determined to be 0.29 eV less for the reaction on Cr_2_CO_2_, and a similar tendency was also attained in Fe_2_C and its O-terminated MXene. As a result, the molecule will likely adhere to Cr_2_C more strongly than its O-terminated surface (Figure 5b). According to Bader charge analyses of the induced net charges on M_2_C MXenes, the electron density distribution between the negatively charged N of NH_3_ and the positively charged Cr/Fe interface would be crucial to the adsorption [55].

Because the highly conductive Ti_3_C_2_T_x_ nanomaterial quickly provides an electron flow to WSe_2_, the Ti_3_C_2_Tx/WSe_2_ band structure, with a partially occupied band spanning the Fermi level, offers a promising catalytic effect for improving sensing processes (Table 2). Due to the electron-deficient nature of adsorbed oxygen species (O_2_ and O) in the fresh air, the electrons were caught by them, resulting in a depletion layer. The ethanol molecules react with the adsorbed active oxygen species to produce volatile gases (H_2_O and CO_2_) and release electrons back into the conduction band when exposed to oxygen-based VOCs (Figure 5c), which reduces the depletion layer and sensor resistance (Ti_3_C_2_T_x_/WSe_2_). Due to the multiple heterojunction interfaces generated, Ti_3_C_2_T_x_/WSe_2_ nanomaterial improves the adsorbed oxygen species (and therefore traps more electrons), which significantly enhances the sensing responsiveness and selectivity for oxygen-containing VOC detection [49].

### 4.2. Application of MXene in Electrochemical Sensors

Over the past few years, electroanalysis has made significant advancements, particularly in the area of electrochemical sensors [68,69]. In general, electrochemical sensors include new photoelectrochemical and electrochemiluminescence sensors, both of which incorporate optical techniques and traditional electrochemical biosensors and non-biosensors. For practical applications, such as non-invasive body fluid monitoring and multiplexed simultaneous detection of disease-related biomarkers, several electrochemical sensing devices have also been created [70,71]. Analytical chemists have been particularly interested in the nanomaterial MXene, which contains transition metal carbonitrides, nitrides, and carbides. These materials have unique structural, electronic, mechanical, thermal, and optical properties, and they are frequently used in electrochemical sensors [70,72,73,74].

The application provides an overview of how MXene-based electrochemical sensor devices are used to identify biomarkers and chemical pollutants (Figure 6a). When MXene is combined with other nanomaterials, the composite may have complementary and synergistic effects that significantly raise the selectivity and sensitivity of electrochemical sensors [75]. A graphitic pencil electrode with perylene diimide (PDI)-MXene integration for dopamine electrochemical sensing has been developed. Due to the efficient electron transfer between PDI and MXene, a nano-adduct with improved electrocatalytic activity and low-energy electronic transitions was produced. The PDI’s anionic groups supported increased active surface area for dopamine’s strong oxidation and specific binding, which resulted in a reduction in applied potential. Meanwhile, strong H-bonding enabled the MXene layers to serve as functionally favorable support for PDI absorption. Due to MXene’s high conductivity, which improved electron transport, the sensor interface’s sensitivity has increased (Figure 6b). A strong DA oxidation was produced by the expansively created nano-adduct, with an ultra-sensitivity of 38.1 μAμM^−1^cm^−2^ and a low detection limit of 240 nM at an oxidation potential of 0.135 V. Additionally, it selectively indicated DA with wide linearity (100–1000 µM) in the absence of physiological interferents. The designed transducing interface has an RSD of 0.1–0.4% and a recovery range of 98.6–100% in human serum samples [76].

Kugalur et al. construct composite biosensors using electrospun one-dimensional (1D) nanofibers of MnMoO_4_ combined with layers of exfoliated 2D MXene for evaluation and simultaneous identification of catechol and hydroquinone in wastewater. The plentiful Faulty edged nanoarchitectures demonstrated electrochemical performance toward the oxidation of catechol and hydroquinone and enhanced the effect of synergy signal amplification. For catechol and hydroquinone, MnMoO_4_-MXene demonstrated oxidation potentials of 0.102 V and 0.203 V, respectively. It had a significant anodic peak current and a distinct and simultaneous detection limit of 0.101 V. The hypothesized 1D and 2D MnMoO_4_-MXene nanomaterial demonstrated a broad linear response for hydroquinone and catechol from 5 nM to 65 nM (Figure 7a). Through using a differential pulse voltametric (DPV) approach with recovery values, this electrode demonstrated high stability of a low limit of detection of 0.26 nM and 0.30 nM for hydroquinone and catechol [77,78]. The carbendazim detection (a pesticide) by MXene-carbon nanohorns-cyclodextrin-metal-organic frameworks (MXene/CNHs/CD/MOFs) was used as a substrate for electrochemical sensors. High adsorption ability for CBZ was made possible by -CD-MOFs, which coupled the host-guest identification of -CD with good pore volume, porosity, and pore structure of MOFs. The MXene-CNHs-CD-MOFs electrode expanded a low detection limit of 1.0 nM and a broad linear range from 3.0–10.0 nM thanks to the synergistic effect of these two materials (Figure 7b). The constructed sensor further shows good repeatability, selectivity, stability, durability, and suitable application [79].

### 4.3. Application of MXene in Biosensors

MXenes have some advantages over previously known 2D nanomaterials, including hydrophilicity, ease of functionalization, high electron transport capacity, a wide range of compositional options, a high surface area as a result of their "accordion-like" biocompatibility and morphology, which makes them desirable for biosensor applications [80,81]. The development of novel bioanalytical platforms is made possible by the functional groups at the surface, such as –F, –OH, and –O, which are suitable for biomolecule functionalization [82,83,84,85].

High conductivity was demonstrated by the nanomaterial made of MXene loaded with MB and AuNPs (MXene-Au-MB). Biomolecules with sulfhydryl termini might be captured by AuNPs, and the MB molecules could then be utilized to provide an electrochemical signal. To create a biosensor, the carboxyl-modified ferrocene (Fc) electrochemical signal sensor was collected by the anchored peptide after the MXene-Au-MB was secured by nafion from the electrode surface. The multifunctional peptide gave the sensing surface the capacity to withstand non-specific adsorption from complicated samples in addition to providing the assaying function. It also contained the antifouling, anchoring, and recognition sequences. The signal of MB in the biosensor system stayed constant as the target concentration increased, whereas the electrochemical signal of Fc steadily declined. The ratiometric detection technique considerably increased the biosensor’s accuracy (Figure 8a–c). The recognizing sequence was identified and cleaved in the presence of a prostate-specific antigen (PSA), model target, and the ratiometric signal of Fc and MB precisely and sensitively identified the intensity of PSA, with a detection ranging from 5–10 ng/mL and a detection limit of 0.83 pg/mL [86].

NIR- and temperature-responsive MXene nanobelt fibers (T-RMFs) with regulated drug delivery and wound-healing potential. The T-RMFs’ superior photothermal activity could more effectively address the challenging requirements for wound healing in clinical practice applications. In addition to giving the T-RMFs good photothermal effects, the MXene nanosheets in these fibrous mats also offered functional groups for cell development. As a temperature-responsive element, the PAAV layer demonstrated outstanding regulated drug release potential and significantly aided stable wound healing over time (Figure 9a–c). Our nanobelts have a great deal of potential for use in the healing process, release of drugs, tissue regeneration, biosensing, and a wide range of other application areas, as evidenced by their simple method, high surface area, wound-healing functions, high mass loading, intriguing nanosheet/nanobelt structure, NIR responsive characteristics of the T-RMFs, and mass manufacturing potential [87].

### 4.4. Application of MXene in Optical Sensors

Their potential for use in optical sensing applications has also attracted much attention due to their distinctive properties, which include superior optical characteristics, excellent hydrophilicity, a large surface area, biocompatibility, ease of surface modification, and high electrical conductivity [88]. Quantum dots, nanosheets, and MXene composites can all be produced from MXenes with the help of adaptable MXene synthesis techniques and the right etching [89,90]. Due to this, optical sensors systems depending on optical sensors made from MXenes have emerged over the past ten years [89,91]. These optical sensors are based on various optical transduction principles, such as surface plasmon resonance, photoluminescence, colorimetry, electrochemiluminescence, and Surface-Enhanced Raman Scattering (SERS) [14,92].

By selectively etching and depositing Ti_3_C_2_T_x_ MXene onto the surfaces of two distinct optical fiber biosensor types, Ti_3_AlC_2_ crystals were converted into this material. An SPR fiber optic sensor receives a 30% increase from the investigation of these two sensors’ improved sensitivity using MXene Ns, while the fiber optic RI sensor receives an improvement of more than eight times. The latter can operate across a wide range with identically excellent performance, while the former can adjust its functioning window to respond to changing detection settings in addition to being an optical fiber RI sensor working material with a broad spectrum that removes the wavelength of optical working restriction for the sensor. Ti_3_C_2_T_x_ MXene is also a fiber optic SPR sensor working window adjustment material that makes it easier to identify these sensors’ spectrum signatures (Figure 10a–c). The ability to easily control the photoelectric characteristics, biochemical processes, and physical stability for use in practice. The benefits of fiber optic sensors and MXene combined include low power usage, small volume, long-distance transmission, anti-electromagnetic interference, and ease of regulation [93].

An easy photoreduction technique is used to increase the SERS activity of MXene and increase the ability to detect antipsychotic drugs. Due to the cooperative action of chemical and electromagnetic mechanisms, MXene anchored with Au NPs causes a spectacular SERS amplification [94,95]. For efficient interaction with probe molecules, the hotspots are created in the nanometer-scale spaces between Au NPs on the TiC surface (Figure 11a). The suggested MXene/Au-NPs SERS platform was used to detect chlorpromazine with a limit of detection of 3.92 × 10^−11^ M and a wide linear range of 10^−1^–10^−10^ M. Additionally, the enhanced TiC/Au-NP SERS effect for the 532 nm excitation shows a better prediction in the order of 109 with relative standard deviations of 13% for uniform and 8.80% for reproducibility [96]. It is possible to tailor the plasma frequencies throughout a wide spectral range, from the mid-infrared to near-infrared, by combining interband transitions and boundary effects (Figure 11b,c). The capacity to create plasmon resonances over a broad range of wavelengths is made possible by the material’s tuneability and layering, which also gives the sensing mechanism more latitude by enabling modulation of the plasma frequency. A typical side-polished optical fiber with MXene coating may stimulate surface plasmon resonances and sustain plasmons in the frequency range used for communications [88].

### 4.5. Application of MXene in Wearable Sensors

Many people are interested in using wearable technology as the interactive platform of the future for robotics, prosthetics, motion detection, and health monitoring [97,98]. These devices have exceptional mechanical compliance and remarkable sensitivity. Beyond the capabilities of standard rigid electronics, flexible electronics require surface-mounted wearable technology to accommodate complex object structures with stable electrical properties during daily movements under cyclic strain situations [99]. The challenges that still exist are (i) the trade-off between mechanical flexibility and electrical performance, (ii) the absence of a fabrication method that is scalable, and (iii) nanomaterials with localized structural surface flaws [2,100,101].

The high-sensitivity, flexible, and biodegradable pressure sensor was created by sandwiching biodegradable polylactic acid thin sheets between interdigitated electrode-coated PLA thin sheets and porous tissue paper that had been MXene-impregnated (Figure 12a,b). The flexible sensing element has a wide range, a low limit of detection (10.2 Pa), a quick response time (11 ms), a low power need (10^−8^ W), reproducibility over 10,000 cycles, and degradability. It is utilized in clinical diagnosis, personal healthcare monitoring, artificial skins, and the prediction of patients’ potential health state, as well as serving as an electronic skin for mapping tactile sensations [102]. A self-powered, flexible, multimodal, MXene-based wearable device is suggested, constructed, characterized, and verified for continuous, real-time physiological biosignal observation. The system includes multipurpose electronics, very sensitive pressure sensors, and power-efficient triboelectric nanogenerators. The main component is 3D-printable MXene, which has distinct electron-deficient and conductive properties (Figure 12c). MXene is joined to a platform called SEBS that resembles skin and has considerable stretchability and positive triboelectric characteristics. This self-powered physiological sensor device allowed for constant radial artery pulse (RAP) waveform observation without the need for independent energy thanks to its sensitivity (6.03 kPa^−1^), power output (816.6 mW m^−2^), the limit of detection (9 Pa), and quick reaction time (80 ms). Near-field communication is used to transmit wireless data and power, as well as its continuous, on-demand, fully self-powered rapid assessment program (RAP) supervision [101].

## 5. Challenges and Prospects

MXenes have been employed in numerous applications in the environment, electronics, and healthcare for the creation of different electronic devices, including energy storage devices, sensors, antennas, nanogenerators, and batteries due to the material’s promising chemical and physical properties [103]. MXene materials have seen significant recent use in sensors and related applications. However, there are still certain drawbacks, such as the difficulty in precisely controlling the homogeneity and thickness of the MXene layer, which has an impact on the sensor’s consistency [93]. Due to their real-time sensing capabilities, portability, flexibility, and reduced electronic waste and environmental effects, flexible and degradable pressure sensors have drawn much attention for potential usage in intelligent robotics, flexible displays, and transient electronic skins [104]. To accomplish full-scale biomonitoring and reduced electronic waste, it is still crucial to concurrently obtain a broad sensing range, robust environmental degradability, high sensitivity, long-term durability, and fast reaction [102,105].

Despite this, there are still some significant obstacles, such as the limitations of MXene sensors and their lack of reproducibility. To obtain thin 2D layers of MXene without undergoing post-processing procedures, a reliable, affordable, and HF-free technique should be created. It is more economical to prevent delamination than to sonicate, which causes surface flaws that impair the characteristics of MXene. MXene’s lengthy shelf life is significantly hampered by its low oxidation stability, which restricts its use in on-site clinical analysis. The transmission of charge between the sheets may be hampered by foreign species used for edge capping, such as ionic ions. The antioxidation properties of MXenes must, therefore, be unlocked by thorough charge transport investigations, protocol optimization, and the invention of new synthesis pathways. Functional groups give MXenes valuable features, including semiconducting behavior and abundant loading of biorecognition components, but it is still difficult to manage their surface chemistry. Realizing their full potential may require improving chemical and thermal stability and synthesizing structures with uniformly terminated or termination-free ends. Bottom-up synthesis techniques can give users improved control over surface flaws, tune active spots for chemical interactions, and manage analyte adsorption. It is currently unclear how MXene surfaces interact with biological molecules on a fundamental level. Their sensing capability can be assessed by measuring the number and orientation of biomolecules on the surface of MXene and by tracking the adsorption kinetics. Utilizing quick and effective biofunctionalization procedures is also crucial [105].

Due to their intriguing properties, including strong metallic conductivity, ease of functionalization, high hydrophilicity, and favorable intercalation characteristics, MXenes are excellent candidates for the construction of sensors. Wearable electrical gadgets and sensors have been created using MXenes. The MXene-based pressure sensors can detect weak or low pressures as well as high-pressure ranges, which makes them useful for gait monitoring applications. Additionally, MXenes are employed to create highly stable, sensitive, and stretchable strain sensors. MXene-based room-temperature gas sensors can be utilized in environmental applications to detect harmful gases or air pollutants, as well as in medical diagnostics to identify biomarkers. The creation of biosensors for the identification of various biomarkers that may be employed for early illness detection or diagnostic reasons also makes use of MXenes. Additionally, low LODs have been seen in the described MXene-based electrochemical devices. Moreover, MXenes with different functional groups are promising candidates for the construction of electrochemical sensors for the specific detection of diverse analytes due to their high conductivity and vast surface area. Additionally, MXenes are an attractive source for biomedical applications due to their great biocompatibility [103].

## 6. Conclusions 

In this review, we have examined the production and use of 2D MXenes and their composites for sensor applications. Both the benefits and the challenges of MXene sensors are thoroughly examined. MXenes have already been utilized to make a variety of sensors, but more study is still required to create new MXenes and enhance the synthesis, manufacturing, and performance of MXene-based sensors. To identify the best option for the mass manufacture of appropriate sensors, alternative MXenes must be researched, as titanium carbide MXene makes up the majority of MXene-based sensors. Researchers will benefit from this review’s insights into the creation of MXene-based sensors for a variety of applications.

## Figures and Tables

**Figure 1 micromachines-14-00247-f001:**
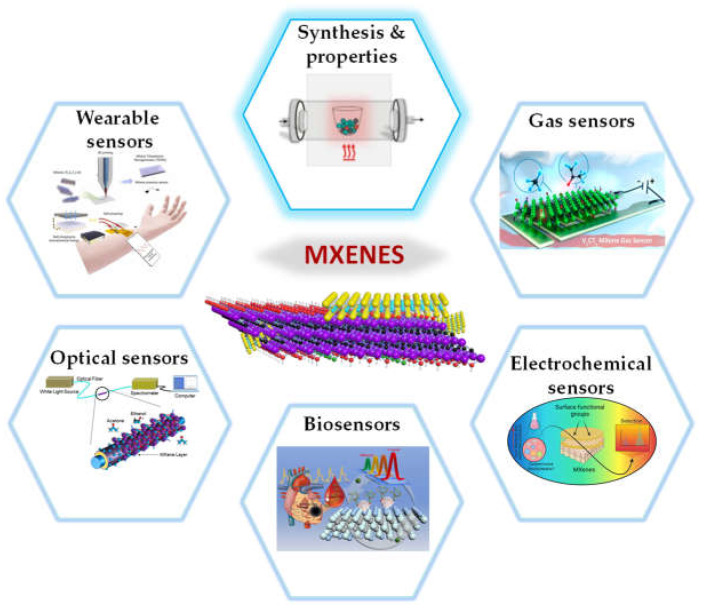
An overview of the applications of MXenes in the sensor, including gas sensors, electrochemical sensors, biosensors, optical sensors, wearable sensors, and synthesis & properties.

**Figure 3 micromachines-14-00247-f003:**
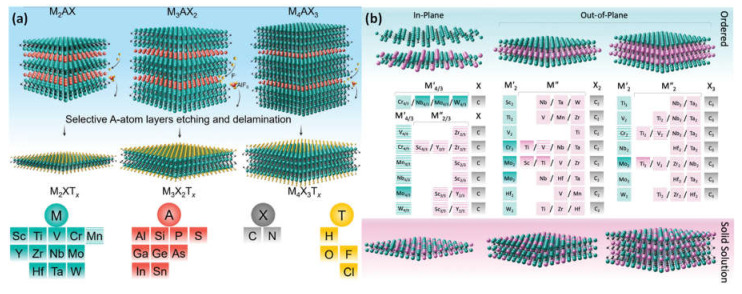
(**a**) After the production of surface terminations (yellow atoms) and the A-group layers were etched selectively (red atoms), three different mono-M MAX phases—M_2_AX, M_3_AX_2_, and M_4_AX_3_—MXenes were produced. M, A, X, and T components that could be present in the MAX and MXene phases; (**b**) the synthesized M_2_X, M_3_X_2_, and M_4_X_3_ MXenes, which are double transition metals (DTM). The transition metals M′ and M′′ are represented by the green and purple elements, correspondingly. Both the in-plane divacancy order (M′_4/3_X) and the in-plane order (M′_4/3_M′′_2/3_X). The order that is out of a plane (M′_2_M′′X_2_ and M′_2_M′′_2_X_3_). M′ and M′′ transition metals are dispersed throughout the disordered MXenes in solid solutions. Additionally, solid solution (Mo_0.8_V_0.2_)_5_C_4_T_x_ was prepared successfully; however, it has not been shown for simplicity’s sake. Reproduced with permission from reference [29].

**Figure 4 micromachines-14-00247-f004:**
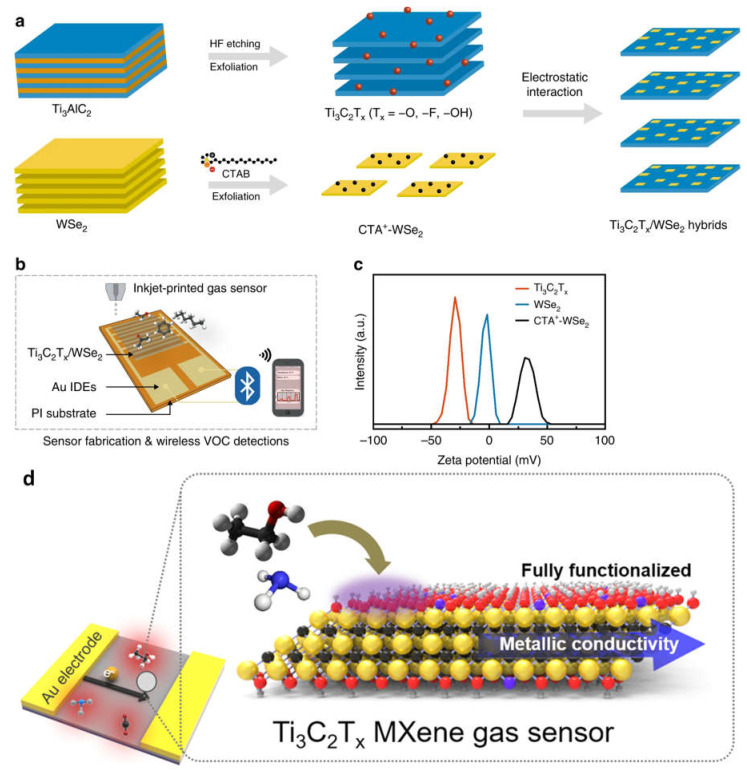
(**a**) Illustration of a schematic of the nanohybrids of Ti_3_C_2_T_x_ and WSe_2_; (**b**) a diagram depiction of inkjet-printed gas sensors used in a wireless monitoring system for detecting volatile organic compounds; (**c**) distributions of the zeta potential for the CTA^+^-WSe_2_ dispersions, WSe_2_, Ti_3_C_2_Tx; (**d**) diagrammatic illustration of the Ti_3_C_2_T_x_ films, its atomic structure. Reproduced with permission from references [49,53].

**Figure 5 micromachines-14-00247-f005:**
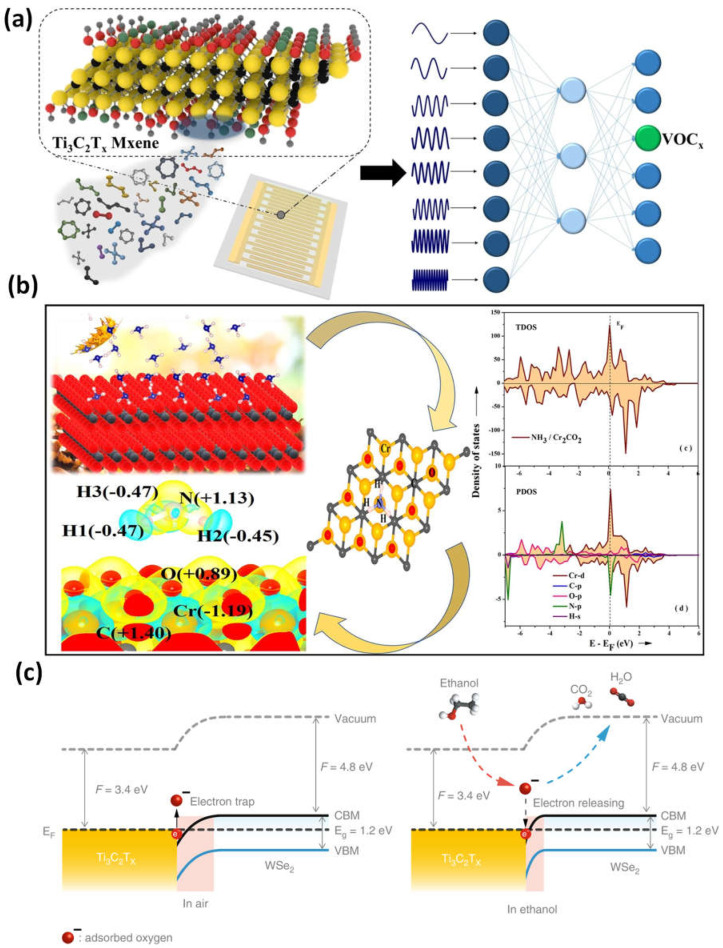
(**a**) The virtual sensor array of MXene 2D material for the ethanol detection in the presence of various concentrations of methanol and water; (**b**) simulation of 2D layered Transition-metal carbide and calculations for NH_3_ gas detection; (**c**) Ti_3_C_2_T_x_/WSe_2_ energy-band graph in air and ethanol, illustrating the change in the depletion region caused by the interaction of adsorbed oxygen species and ethanol molecules. Reproduced with permission from references [49,54,55].

**Figure 6 micromachines-14-00247-f006:**
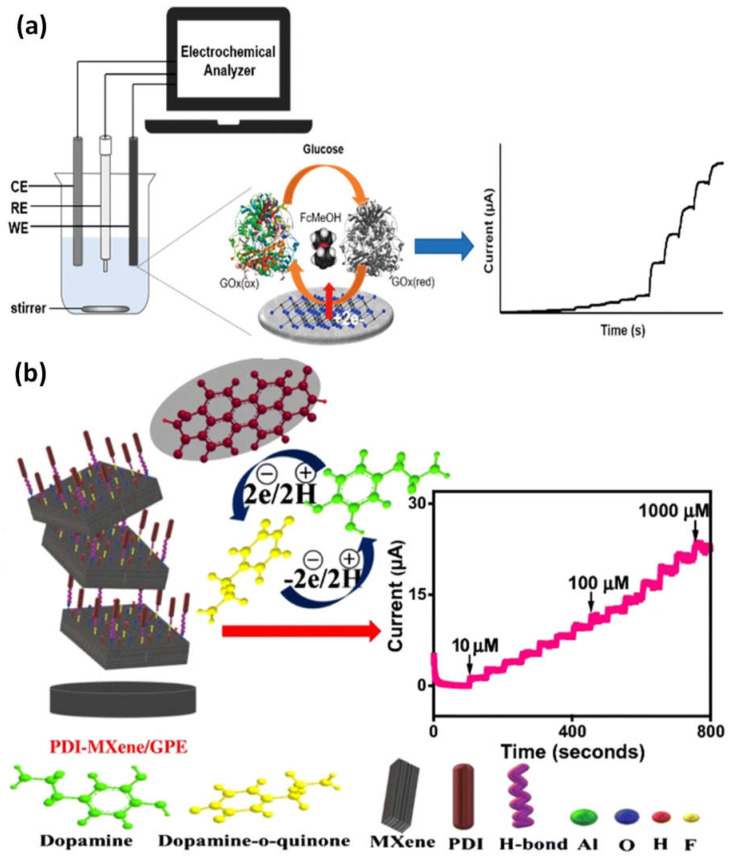
(**a**) An example of a diagram showing how the electrochemical glucose sensor functions; (**b**) a graphitic pencil electrode with perylene diimide-MXene integration for the electrochemical sensing of dopamine and amperometric current-time response curves for the gradual addition of various concentrations of dopamine solution. Reproduced with permission from references [75,76].

**Figure 7 micromachines-14-00247-f007:**
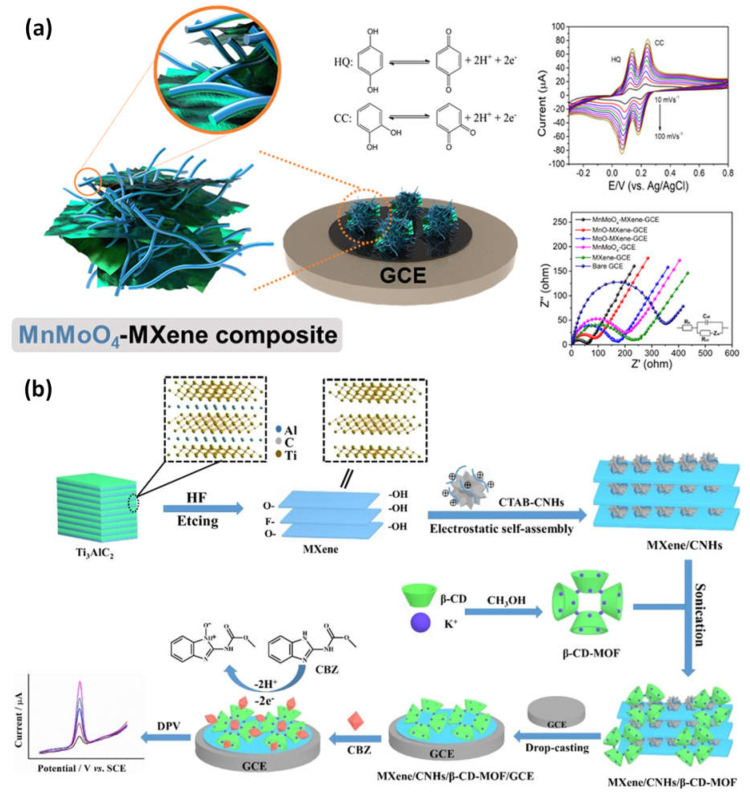
(**a**) Examples of the production of a 1D/2D MnMoO_4_/MXene nanomaterial sensing for the simultaneous detection of catechol (CC) and hydroquinone (HQ), as well as EIS and DPV responses to different concentrations of HQ and CC in MXene nanocomposite materials; (**b**) based on the MXene-CNHs-CD-MOF electrode, an electrochemical sensor for the detection of carbendazim was developed. Reproduced with permission from references [77,79].

**Figure 8 micromachines-14-00247-f008:**
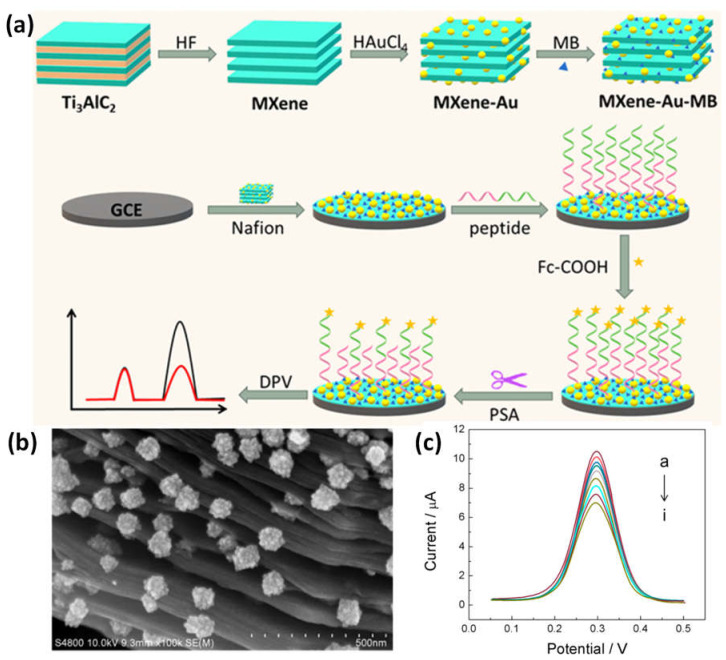
Diagrammatic representation of the operation of a ratiometric antifouling electrochemical biosensor is indicated in (**a**), along with (**b**) SEM pictures of a composite made of MXene and Au at various magnifications and (**c**) a sensor interface that reacts to PSA’s DPV signal without a standard solution. Reproduced with permission from reference [86].

**Figure 9 micromachines-14-00247-f009:**
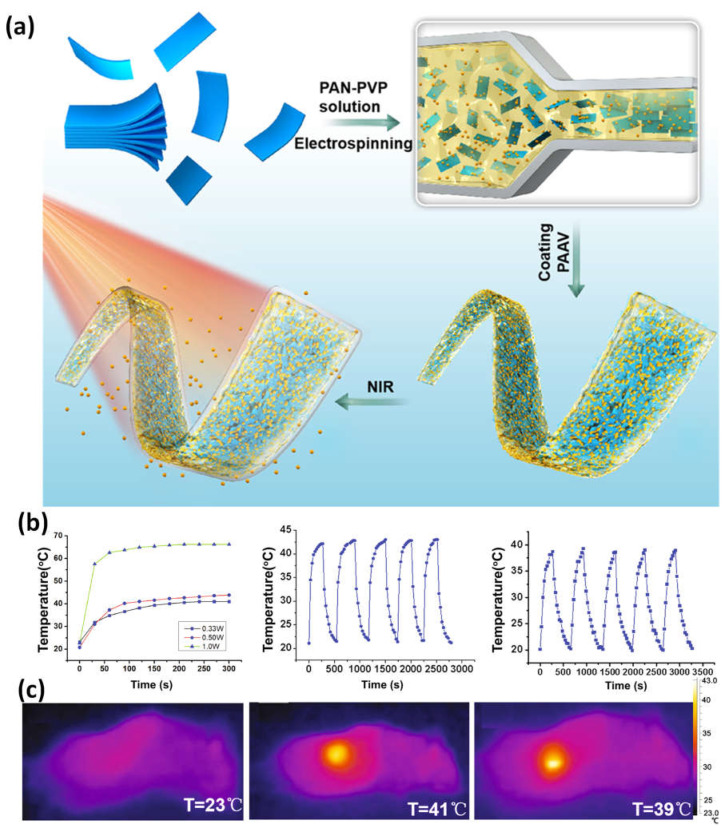
(**a**) A diagram showing how T-RMFs nanobelts are made and coated; (**b**) the RMFs’ temperature growth profiles under NIR light (0.33, 0.50, 1.0 W), as well as the T-RMFs’ and RMFs’ temperatures over five on/off cycles; (**c**) thermal pictures taken with NIR light (0.33 W) of the comparison group, T-RMFs, and RMFs. Reproduced with permission from reference [87].

**Figure 10 micromachines-14-00247-f010:**
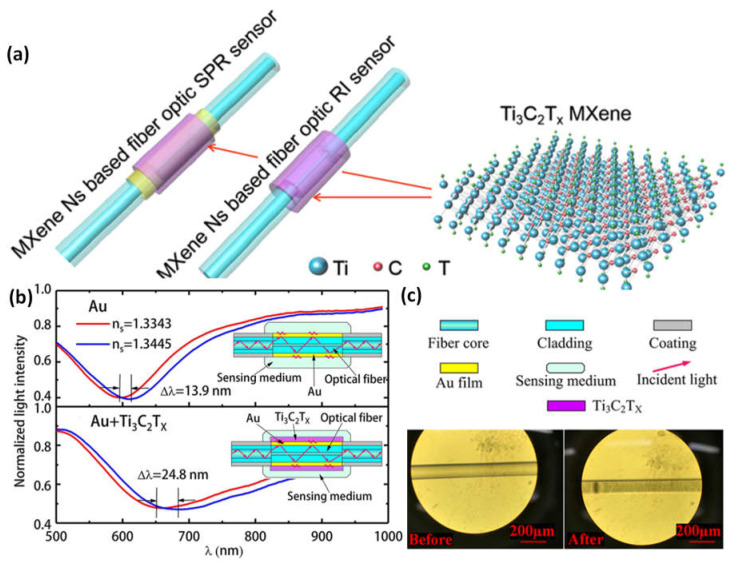
(**a**) 2D MXene Ti_3_C_2_T_x_ improves the sensitivity of optical fiber nanosensors, including RI and SPR sensors; (**b**) differences in the standardized intensity of light to wavelength for the traditional fiber optic surface plasmon resonance (SPR) nanosensor of Au film and the optical sensor with the MXene layer; (**c**) The graphs of an optical fiber RI detector before and after the MXene Ns modification. Reproduced with permission from reference [93].

**Figure 11 micromachines-14-00247-f011:**
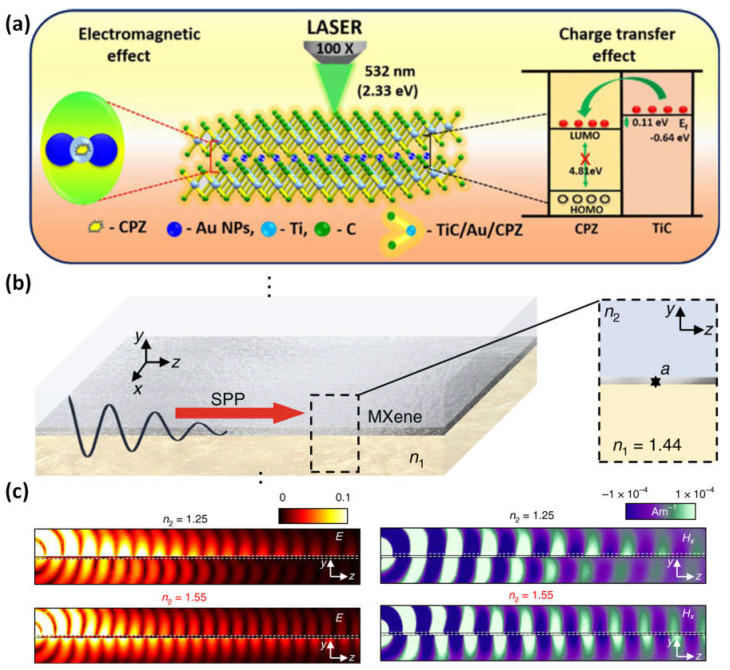
(**a**) The simple photoreduction procedure used to incorporate gold nanoparticles into MXene to increase its SERS activity to detect antipsychotic drugs; (**b**) a sketch structure of MXene, which consists of two semi-infinite media with the refractive indices n_1_ (the bottom material, SiO_2_) and n_2_ (the top material), and Ti_3_C_2_T_x_ sandwiched between two thin layers; (**c**) numerical findings for the distributions of Hx field and electric field on the yz planes, respectively, taking into account several materials for the top dielectric: n_2_ = 1.25 (top) and n_2_ = 1.55 (bottom). Reproduced with permission from references [88,96].

**Figure 12 micromachines-14-00247-f012:**
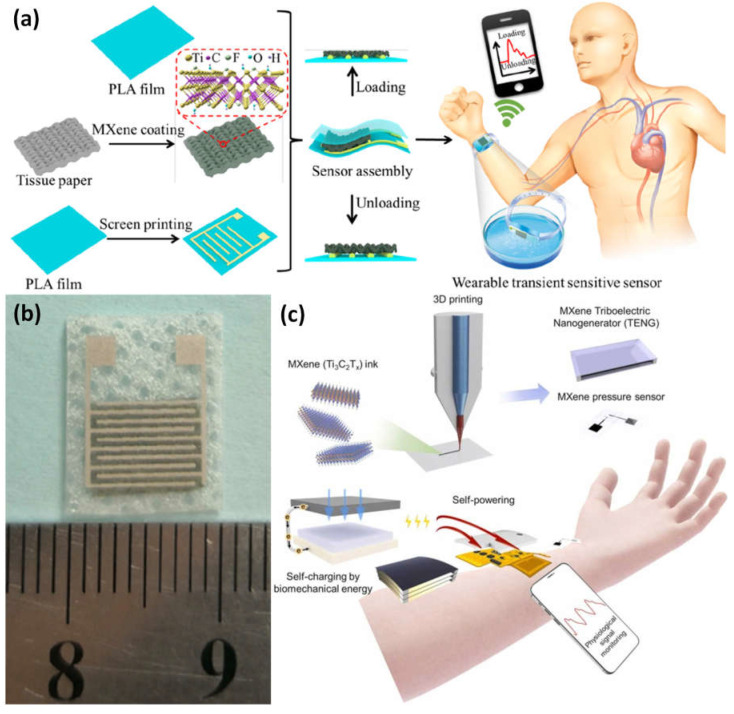
(**a**) Sandwiching a thin sheet of a biodegradable polylactic acid (PLA) and an interdigitated electrode-coated PLA between two porous MXene-impregnated tissue papers to anticipate the potential health condition of patients and serve as a tactile input mapping electronic skin; (**b**) images of the stretchable transient pressure sensor for wearables; (**c**) the suggested, designed, characterized, and verified wearable device for MXene-based, adaptable, and integrated physiological biosignals monitoring for continuous, real-time. Reproduced with permission from references [101,102].

**Table 1 micromachines-14-00247-t001:** Comparison of the physicochemical properties of MXenes with other nanomaterials.

Type of Nanomaterial	Conductivity (S/cm)	Surface Area (m^2^ g^−1^)	Band Gap (eV)	Biocompatibility	Ref.
MXene (Ti_3_C_2_Tx)	9050	97.5	1.75	biocompatibility	[37]
MoS_2_	728.2	27.82	1.80	-	[38]
Multi-Walled Carbon Nanotube (MWCNT)	640	280	1.82	biocompatibility	[39]
Graphene	105	2630	0.952	biocompatibility	[40]
SWCNT	283	1315	0.9	biocompatibility	[41]
ZIF8 (Metal-organic framework)	0.5 × 10^−3^	2170	5.3	biocompatibility	[7]
Ni-COF (Covalent organic framework)	1.2	120.5	3.84	-	[42]
SiO_2_	5 × 10^−12^	640	9.3	biocompatibility	[5]
C_3_N_4_	0.104	142.1	2.9	-	[43]

**Table 2 micromachines-14-00247-t002:** Modification of MXene materials for gas sensing performances.

Mxene Materials	Detection Concentration (ppm)	Detection Gas	Response/Recovery Time (s/s)	Detection Temperature (°C)	Ref.
Ti_3_C_2_T_x_/CuO	50	Toluene	270/10	250	[56]
Ti_3_C_2_T_x_/Pd	4000	H_2_	37/161	RT	[57]
Ti_3_C_2_T_x_/ZnO	100	NO_2_	34/105	RT	[50]
Ti_3_C_2_T_x_/SnS_2_	1000	NO_2_	64/110	RT	[58]
Ti_3_C_2_T_x_/Co_3_O_4_	10	HCHO	83/5	RT	[59]
Ti_3_C_2_T_x_/GO	100	NH_3_	-	RT	[60]
Ti_3_C_2_T_x_/PEDOT:PSS	100	NH_3_	116/40	RT	[61]
Ti_3_C_2_T_x_/SnO_2_	50	NH_3_	36/44	RT	[62]
Ti_3_C_2_T_x_/rGO/CuO	100	Acetone	6.5/7.5	RT	[63]
Ti_3_C_2_T_x_/Fe_2_(MoO_4_)_3_	100	N-butane	18/24	RT	[64]
Ti_3_C_2_T_x_/Co_3_O_4_@PEI	100	NO*_x_*	1.6/73.1	RT	[65]
Ti_3_C_2_T_x_/In_2_O_3_	200	Ethanol	0.4/0.5	RT	[66]
Ti_3_C_2_T_x_/W_18_O_49_	0.17	Acetone	5.6/6	300	[67]
Ti_3_C_2_T_x_/Pd	4000	H_2_	37/161	RT	[57]
Ti_3_C_2_T_x_/WSe_2_	40	Ethanol	9.7/6.6	RT	[49]

## Data Availability

Not applicable.

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
