# Peer review of "Application Prospects of MXenes Materials Modifications for Sensors"

_micromachines, 2023, doi:10.3390/mi14020247_

Round 1

Reviewer 1 Report

In this manuscript, the authors studied different synthesis procedures of MXenes, its structures and analyzes their physical and chemical properties, with a focus on optical, electronic, structural, and mechanical characteristics. On the basis of these, the authors concentrated on application of Mxenes material for sensors. I am favorable to the publication of this manuscript, and I have several comments:

1. Some words can be improved/corrected, name a few examples here:

- In abstract, "development of MXenebased sensor are analysis"->"development of MXenebased sensor are analysed"

- In 1st paragraph of Introduction, "including appropriate conductivity on par with metals"->"including conductivity comparable to that of metals"

- In 3rd paragraph of Introduction, "Due to the rapid advancement of information technologies and electronics" -> "Due to the rapid development of information technologies and electronics"

2. The analysis of Mxenes properties is too short. It would be more complete if authors can provide more content, like band structure for electronic properties.

Author Response

Micromachines

Manuscript ID: micromachines-2174069

Type of manuscript: Review

Title: Application Prospects of Mxenes Materials Modifications for Sensors

Dear Editor,

We appreciate your consideration of our manuscript for publication in Micromachines. We also want to express our gratitude to the reviewers for their valuable suggestions and comments.

In the revised version, which is being submitted along with the response letter, we incorporate all the suggestions requested by the reviewers. A detailed summary of our responses is provided on the following pages. We hope that the revised manuscript is now suitable for publication in Micromachines.

Best regards,

Dr. Tran Anh Vy

Institute of Applied Technology and Sustainable Development

Nguyen Tat Thanh University, Ho Chi Minh city, Viet Nam

Email: tavy@ntt.edu.vn,  tranhvy@gmail.com

Reviewer #1

In this manuscript, the authors studied different synthesis procedures of MXenes, its structures and analyzes their physical and chemical properties, with a focus on optical, electronic, structural, and mechanical characteristics. On the basis of these, the authors concentrated on application of Mxenes material for sensors. I am favorable to the publication of this manuscript, and I have several comments:

  1. Some words can be improved/corrected, name a few examples here:

- In abstract, "development of MXenebased sensor are analysis"->"development of MXenebased sensor are analysed"

- In 1st paragraph of Introduction, "including appropriate conductivity on par with metals"->"including conductivity comparable to that of metals"

- In 3rd paragraph of Introduction, "Due to the rapid advancement of information technologies and electronics" -> "Due to the rapid development of information technologies and electronics"

Response:

We've re-checked the errors as mentioned by the reviewer, and re-checked the manuscript as a whole and corrected them. We appreciate the reviwer's meticulous reviews.

  1. The analysis of Mxenes properties is too short. It would be more complete if authors can provide more content, like band structure for electronic properties.

Response:

We have added more content to Mxenes properties section. The "band structure for electronic properties" is displayed in the revised manuscript. I hope these changes will satisfy the reviewer's request.

Reviewer 2 Report

This is  an excellent review. I suggest to authors to review also done regarding electromagnetic shielding with  MXenes in microwaves since very good results were obtained.

Author Response

Micromachines

Manuscript ID: micromachines-2174069

Type of manuscript: Review

Title: Application Prospects of Mxenes Materials Modifications for Sensors

Dear Editor,

We appreciate your consideration of our manuscript for publication in Micromachines. We also want to express our gratitude to the reviewers for their valuable suggestions and comments.

In the revised version, which is being submitted along with the response letter, we incorporate all the suggestions requested by the reviewers. A detailed summary of our responses is provided on the following pages. We hope that the revised manuscript is now suitable for publication in Micromachines.

Best regards,

Dr. Tran Anh Vy

Institute of Applied Technology and Sustainable Development

Nguyen Tat Thanh University, Ho Chi Minh city, Viet Nam

Email: tavy@ntt.edu.vn,  tranhvy@gmail.com

Review #2

This is  an excellent review. I suggest to authors to review also done regarding electromagnetic shielding with  MXenes in microwaves since very good results were obtained.

Response:

Thanks to the reviewer's positive comments, we checked out this study. In this review, we focus on mentioning and in-depth analysis of the sensor by Mxene material. We also considered another use of Mxene, electromagnetic shielding in microwaves, in the Introduction. Hopefully in the near future we will focus on further research on electromagnetic shielding in microwaves by Mxens materials.
